# Automated Remote Detection of Falls Using Direct Reconstruction of Optical Flow Principal Motion Parameters

**DOI:** 10.3390/s25185678

**Published:** 2025-09-11

**Authors:** Simeon Karpuzov, Stiliyan Kalitzin, Olga Georgieva, Alex Trifonov, Tervel Stoyanov, George Petkov

**Affiliations:** 1GATE Institute, Sofia University, 1164 Sofia, Bulgaria; simeon.karpuzov@gate-ai.eu (S.K.); olga.georgieva@gate-ai.eu (O.G.); aleks.trifonov@gate-ai.eu (A.T.); tervel.stoyanov@gate-ai.eu (T.S.); 2Stichting Epilepsie Instellingen Nederland (SEIN), Achterweg 5, 2103 SW Heemstede, The Netherlands; skalitzin@sein.nl; 3Image Sciences Institute, University Medical Center Utrecht, Heidelberglaan 100, 3584 CX Utrecht, The Netherlands; 4Faculty of Mathematics and Infromatics, Sofia University, St. Kliment Ohridski, 1164 Sofia, Bulgaria

**Keywords:** fall detection, optical flow, real-time detection, video-surveillance, principal motion parameters

## Abstract

**Highlights:**

**What are the main findings?**
Falls can be reliably detected using optical flow video processing algorithms.Real-time performance is enhanced by direct reconstruction of principal motion parameters.

**What is the implication of the main finding?**
The proposed algorithm allows for modular integration into existing patient care observation systems.It provides non-obstructive, maintenance-free, and privacy-respecting tools for safety.

**Abstract:**

Detecting and alerting for falls is a crucial component of both healthcare and assistive technologies. Wearable devices are vulnerable to damage and require regular inspection and maintenance. Manned video surveillance avoids these problems, but it involves constant labor-intensive attention and, in most cases, may interfere with the privacy of the observed individuals. To address this issue, in this work we introduce and evaluate a novel approach for fully automated fall detection. The presented technique uses direct reconstruction of principal motion parameters, avoiding the computationally expensive full optical flow reconstruction and still providing relevant descriptors for accurate detections. Our method is systematically compared with state-of-the-art techniques. Comparisons of detection accuracy, computational efficiency, and suitability for real-time applications are presented. Experimental results demonstrate notable improvements in accuracy while maintaining a lower computational cost compared to traditional methods, making our approach highly adaptable for real-world deployment. The findings highlight the robustness and universality of our model, suggesting its potential for integration into broader surveillance technologies. Future directions for development will include optimization for resource-constrained environments and deep learning enhancements to refine detection precision.

## 1. Introduction

### 1.1. Motivation

Detection of falls is an increasingly important task in the modern world. Approximately one-third of adults aged 65 and above in the European Union experience one fall annually [1]. Falls are dangerous and often lead to serious injuries and health complications. In many cases, falling can be fatal [2]. Falls frequently require medical attention, and people often need to be hospitalized [2]. This rounds up to an annual cost of treatment around €25 billion, which is a substantial toll [3]. Naturally, developing alerting and prevention strategies, protocols, methods, and technologies is an important endeavor that helps reduce the negative outcomes and costs related to falls in the elderly population as well as in specific groups of vulnerable individuals such as epileptic patients.

There are different approaches for automated fall detection that are used in practice. Frequently used devices include wearable sensors and ambient devices. Wearable sensors, which are the most accurate [4,5,6], are widely used. They offer round the clock monitoring and instant response in the event of an emergency. Wearable sensors are affordable and can be customized based on the individual using them. Ambient sensors [7] also offer substantial benefits—they require no compliance from the user, provide broad area coverage, and are usable continuously. In the current work, we use a visual-based sensor (camera) for data collection, specifically for the purpose of fall detection. When compared to wearables, camera-based systems offer several advantages. They can successfully identify falls in any individual present within the monitored space. Wearable devices presuppose that their user is already at risk of falling. Camera-based systems have no such assumption, thus broadening their field of application. Camera infrastructure is more robust and, once installed, requires little support and/or maintenance. Wearable devices need to be charged and placed properly. They may also pose certain issues of compromising comfort, privacy, and add stigmatizing effect for the individual wearing them. Remote sensors, such as cameras, will monitor and alert without obstructive effects. When compared to ambient devices, cameras are less expensive and require less maintenance while offering comparable benefits in terms of detection accuracy. There are several disadvantages when using cameras that are worth mentioning. They may be less accurate in certain scenarios due to unfavorable lighting conditions. Obstructions and limited coverage my further impact detection capability. Personal privacy is also impacted when working with video data. These are important points that need to be accounted for when selecting a sensor for the task.

Our method for fall detection works by processing video data with Optical Flow (OF) [8,9,10,11] techniques. Such methods are widely used in different computer vision tasks, and their application for fall detection can offer various benefits. OF methods allow us to capture motion without the need for physical markers or wearable devices [12]. If certain conditions are met, of methods allow for real-time analysis of movement [13]. To work properly, Optical Flow methods only require simple and cheap hardware such as a basic USB camera, connected to a personal computer [14]. OF techniques have been shown to perform well under a dynamic environment—complex scenes with multiple moving objects [15]. Finally, Optical Flow methods can be combined with machine learning algorithms [16] in order to better solve different computer vision tasks.

### 1.2. Related Works

Fall detection using camera-based systems is an increasingly important topic in the field of computer vision and digital healthcare. A lot of work is constantly being performed in the field. Many different approaches are available; we refer the reader to the relevant literature [4,17]. Here we will go over only those methods that use Optical Flow evaluation in their workflow. In [18], the motion vector field for each two consecutive frames is calculated. The mean value of the vertical component of the vector field for each two frames is evaluated. From it, certain features such as the maximum vertical velocity and acceleration are derived, and together with the maximum amplitude of the recorded sound, a feature vector is defined. An SVM [19] classifier is trained on a large dataset, and subsequently the classifier is evaluated on new data. For this work, both video and audio data are used. The method proposed in [20] utilizes a mixture of background subtraction [21], standard Optical Flow techniques, and Kalman filtering [22] to detect falling events. A feature vector that consists of the “angle”, the “ratio” (defined as width-to-height relationship of a bounding rectangle), and the “ratio derivative” (defined as the velocity with which the silhouette changes) is used with a k-Nearest Neighbor [23] classifier to classify whether to observed movement event is a fall. In [24], OF is calculated, and various features of points of interest are derived. These are then used by a Convolutional Neural network [25] for classification of falling events. The system also includes a two-step rule-based motion detection system that handles large, sudden movements and applies rule-based mechanisms when variations in optical flow exceed a threshold.

In the current paper, Optical Flow is calculated by a novel technique developed in our group [9]. This OF method provides substantial benefits when compared to traditional OF algorithms. GLORIA is a group parameter reconstruction method known for its computational efficiency and speed. In the following sections, we show how to utilize the benefits of this method in solving the fall detection problem. We would like to point out that the method for fall detection we are presenting has been designed and tested on indoor data with a single moving person in frame but can certainly be applied in open space applications as well. We would also note that, even though we analyze video data, our method is set up in such a way that respects the privacy of the individual being filmed. This is a specific property of the GLORIA algorithm—it extracts only principal motion information from a video. Any other structure is lost, and the results of the technique cannot be used for any other purposes (unethical or otherwise). This is a central theme in computer vision tasks. We have gone in more detail in a previous work of ours [13] regarding sensitivity of the video data. Our technique has low computational requirements. It can run in real-time on low to mid-range CPUs. This is a significant advantage when compared to other contemporary methods which require a dedicated GPU for their detection workflow.

### 1.3. Organization

This paper is organized in the following way: Our novel fall detection method is introduced in the following chapter. We present its description, the machine learning methods that are used for classification, the evaluation metrics, and the video datasets on which our algorithm is evaluated. Subsequently, we present our results on said datasets and provide comments about the method based on accuracy, speed, and other conditions such as camera placement. In the final section of the paper, we go over further comments, directions for future improvements, limitations, and the properties of our approach that make it stand out.

## 2. Materials and Methods

Our method consists of three steps. First, the GLORIA technique allows us to extract motion features from a subsection of a video. We use it to calculate principal motion information (two translations, two shears, dilatation, and rotation) and organize this data into a lower-dimensional structure—a 6 × 150 vector. This data is then used as an input to our CNN. The network classifies events based on that input, and subsequently, the LSTM allows us to better capture short-term dynamics (a key property of these types of methods). The final step increases detection accuracy. We go over each step in more detail in the following subchapters, starting with a brief description of the GLORIA technique.

### 2.1. GLORIA Net

Description of our current method begins by introducing the OF problem in Equation (1). We refer to an individual pixel in a color image frame as $L^{c}(x,y,t)$. Here, we denote with $\left(x,y,t\right)$ the spatial coordinates and time, and c is an index for the color channel (RGB). If we assume that all temporal changes in the image content arise solely from scene deformation, and we define the local velocity vector field as v(x,y,t), then the corresponding image transformation can be expressed as:(1)$$\frac{dL^{c}}{dt}=-\nabla L^{c}\cdot v\equiv \nabla_{v}L^{c}$$

In Equation (1), by $\nabla_{v}$ we denote the vector field operator, $c=1:N_{c}$ is the current color channel, $N_{c}$ is the total number of color channels, and t is the time. The velocity field can account for a wide variety of objects’ basic motions, such as rotations, dilatations, translations, etc. For the current method, however, we do not need to calculate the velocity vector field for each point, as we can directly reconstruct the global features of the optic flow by considering only specific aggregated values associated with it. Specifically, we aim to identify the global two-dimensional linear non-homogeneous transformations, which comprise rotations, translations, dilatations, and shear transformations that characterize fall movements. For this purpose, we use the GLORIA algorithm. The vector field operator from Equation (1) takes the following form:(2)$$\nabla_{v}\equiv v\cdot \nabla \equiv \sum_{k}v_{k}\nabla_{k};\;\nabla_{k}L^{c}\equiv \frac{\partial L^{c}}{\partial x_{k}}$$

The representation in Equation (2) is used for the decomposition of the vector field v  into a group of several known transformations. In Equation (3), we denote by $v^{u}$ the transformation generators within the group and by $A^{u}$ their corresponding parameters.(3)$$v\equiv \sum_{u}A^{u}v^{u}$$

With Equation (3), we introduce a set of differential operators for the group of transformations that form a Lie algebra:(4)$$G^{u}\equiv \sum_{k}v_{k}^{u}\nabla_{k}$$

We can apply Equation (4) to the group of six general linear non-homogeneous transformations in two-dimensional images to derive the global motion operators:(5)$$G^{translation_{x}}=\nabla_{x};\;G^{translation_{y}}=\nabla_{y};\;{\;G}^{dilation}=x\nabla_{x}+y\nabla_{y};\;{\;G}^{rotation}=y\nabla_{x}-x\nabla_{y};\;G^{shear_{1}}=x\nabla_{y};\;G^{shear_{2}}=y\nabla_{x}\;$$

The aggregated values related to the translations, shears, rotation, and dilatation $A_{u}$ can be expressed through the structural tensor $S_{kj}$ and the driving vector field $H_{k}$:(6)$$H_{k}=\;-\sum_{c}\frac{dL^{c}\left(x,t\right)}{dt}\nabla_{k}L^{c}\;S_{kj}=\;\sum_{c}\nabla_{k}L^{c}\nabla_{j}L^{c};j,k=1,2\;A^{u}={\left(S^{uv}\right)}^{-1}H^{u}$$

The GLORIA algorithm allows us to skip pointwise OF calculation and instead obtain a 6-element mean-flow vector for each two consecutive frames. This is very advantageous as it allows us to speed up the process significantly while preserving a high degree of accuracy. Another strength of this method is that it works regardless of camera position. It is also important to note that GLORIA can work with spectral data (in our case the color images that make up the video). As seen from Equation (1), all spectral channels contribute to the solution of the inverse problem. It is in general, and especially in the case of using only intensity images, an underdetermined problem because the local velocities in the directions of constant intensity can be arbitrary. This degeneracy is less likely to occur in multichannel images.

We examine the videos by splitting them into *N* = 150 frame subdivisions. For each subsection, we have a 6 × 150 array. We use our datasets to train a Convolutional Neural Network (CNN) that works with our GLORIA vectors and subsequently classify whether new video data contains a fall event or not.

The CNN uses a 1D-like architecture by applying 2D convolutional layers, allowing it to capture patterns across the width while preserving vertical structure. The network has an input layer tailored to arrays of size 6 × 150. It uses three convolutional blocks to extract features. The first two blocks each have a convolutional layer, a batch normalization, ReLU activation function, and max pooling step. These layers extract different features and reduce the width of the feature maps. The last convolutional block is used to refine finer details and features and can be optional. Afterwards, the feature maps are flattened into a one-dimensional vector, passed through a dense layer with 64 neurons, and regularized with dropout step to prevent overfitting. The final fully connected layer has two neurons that are used for binary classification, followed by a *softmax* and classification layer to output probabilities and compute the cross-entropy loss. The model is trained using an *Adam* optimizer with a learning rate of 0.001, mini-batches of size 32, and for up to 16 epochs. Validation is performed regularly using test data, and L2 regularization (0.001) is applied to improve generalization. This architecture is specially designed for our type of structured temporal data. Our main idea with this approach is that a fall is a very fast and abrupt event that will have an effect on all six elements of the flow vector. The CNN helps us to additionally refine and classify features that reflect the fall. A diagram of the network’s layers is presented in Figure 1:

### 2.2. LSTM

The GLORIA algorithm generates a sequence of 6 × 1 vectors, each quantifying the total motion between consecutive video frames, thereby encoding inherent temporal dependencies. To classify videos for fall detection, a model capable of processing this sequential data while preserving historical context was essential. Traditional recurrent neural networks (RNNs) are fundamentally limited in maintaining long-term contextual information. Consequently, we adopted Long Short-Term Memory (LSTM) networks [26], a robust extension designed to overcome these limitations through its specialized gating mechanism. Despite their prevalent use in natural language processing, LSTMs have demonstrated strong performance in time-series analysis, suggesting their utility for our classification problem. Our solution employs a hybrid CNN-LSTM architecture to leverage the respective strengths of both models. Their specific properties allow them to capture short-term dynamics like sudden movements and long-term tendencies like slow postural movements. This makes them very effective for temporal data such as human activity or in our current case—fall detection. Convolutional Neural Network (CNN) layers are first utilized to extract local patterns and refine the raw motion vector representation. The output from these CNN layers is then fed into LSTM layers, which process the refined vectors sequentially to identify meaningful temporal features. This integrated approach effectively combines the spatial feature extraction capabilities of CNNs with the temporal modeling power of LSTMs, resulting in enhanced classification accuracy for fall detection. This LSTM approach allows us to additionally increase the accuracy of our method. Figure 2 shows where the Bidirectional LSTM block fits in our CNN layer structure:

### 2.3. Datasets

We have used a total of three datasets for method training and evaluations in this work. They are all public and available for download by anyone. We have decided to use widely popular and freely available resources as we believe it is a good way to benchmark our method’s capabilities.

#### 2.3.1. UP Fall Detection Dataset

This dataset [27] contains 1118 videos in total with FPS of 18 frames per second. There are two camera positions from which falls are recorded. Videos are short, with an average duration of roughly 16 s. The camera is placed horizontally with respect to the floor. Frame size is 640 × 480. The recorded video is in color. The dataset is labeled by activity. These are the motions the recorded person is performing—jumping, laying on the ground, sitting in a chair, walking, squatting, falling, and standing still upright.

#### 2.3.2. LE2I Video Dataset

This dataset [28] contains 191 videos with FPS of 25 frames per second. There are numerous camera positions, including camera placement in one of the upper corners of the room. This means that sometimes, depending on fall direction and camera placement, the velocities of the person related to the fall will not have a significant y-component. Frame resolution is 320 × 240. Here the videos are labeled by marking during which frames of the video a fall occurs.

#### 2.3.3. UR Fall Detection Dataset

This dataset [29] contains 70 videos with FPS of 30 frames per second. Like the previous dataset, camera positions are numerous. Kinect cameras are used to record fall events. The videos are in color. We use this dataset mainly as out-of-distribution set to further evaluate the performance of the GLORIA Net method.

### 2.4. Evaluation Parameters

In the current work, we are interested in binary classification. In order to evaluate the performance of our models, we introduce a number of statistical values, derived from the confusion matrix. This is a 2 × 2 matrix that is used to compare ground-truth labels to labels predicted from our models. It has the following entries:True positives (TP)—number of times we have correctly predicted an event as positive (e.g., our model predicts a fall, and a fall occurred indeed);False positives (FP)—number of times we have incorrectly predicted an event as positive (Type I error);True negative (TN)—number of times we have correctly predicted an event as negative (e.g., our model predicts no fall, and a fall did not occur);False positives (FN)—number of times we have incorrectly predicted an event as negative (Type II error).

Using these values, we can define the following measures:(7)$$Accuracy=\;\frac{TP+TN}{TP+TN+FP+FN}$$

*Accuracy* is a useful evaluation parameter as it shows us the overall proportion of correctly predicted events. Another measure we use in the article is the *Sensitivity*:(8)$$Sensitivity=\;\frac{TP}{TP+FN}$$

This measure indicates how often the model correctly classifies positive cases. Similarly, the *Specificity* shows us how good the model’s prediction is for negative cases:(9)$$Specificity=\;\frac{TN}{TN+FP}$$

*Precision* shows us the rate of predicted positives that are classified correctly:(10)$$Precision=\;\frac{TP}{TP+FP}$$

The *F*1-*score* is a good example for a single parameter that can evaluate false positives and false negatives at the same time:(11)$$F1=\;2\frac{Precision\cdot Sensitivity}{Precision+Sensitivity}$$

In order to demonstrate the performance of our model graphically, we will use Receiver Operating Characteristic (ROC) curves. They plot the *Sensitivity* (*True Positive Rate*) against the *False positive rate* (1 − *Specificity*) at different thresholds. They are useful as the Area Under the Curve (AUC) can be used to describe how effective a model is. The closer AUC is to 1, the better the model is. An AUC value close to 0.5 indicates random guessing. The introduced values so far are dimensionless and indicate a percentage.

Used together, all the measures described so far can give a very detailed evaluation of the performance of our predictive model.

## 3. Results

A Lenovo^®^ ThinkPad (Lenovo, Hong Kong) with an Intel^®^ Core i5 CPU (Intel, Santa Clara, CA, USA) and 16 Gb of RAM is used to process videos from our sets. Algorithm realization is carried out in MATLAB^®^ R2023b environment.

### 3.1. GLORIA Net Results

For training and evaluation of this method, we combined the UP-Fall dataset and the LE2I dataset. The CNN was trained on 90% of the data (as well as finetuning the hyperparameters and other network-related settings). Performance evaluation was then carried out on the remaining 10% of the data. We also introduced an out-of-distribution dataset (UR Fall Detection Dataset), in order to give a more detailed and clearer picture on model performance as sometimes CNNs tend to overfit on their training data. This also helps us to verify the quality, adequacy, reusability, and robustness of the selected features. ROC curves for GLORIA Net are available in Figure 3:

The other evaluation measures, introduced in Section 2.4, are displayed in Table 1:

With an included LSTM approach, the performance of our method is slightly improved (we register less false negative (FN) events). In this case, the same evaluation measures are presented in Table 2:

The results in Table 2 show the beneficial effect of the LSTM approach.

### 3.2. Comparison in Processing Times

We also provide a detailed evaluation of the speed with which we process the video data and classify events in a 150-frame window extract from the video. Figure 4a shows us the time our method needs to decide whether a fall event is present or not in a 150-frame time window for different camera resolutions. Figure 4b displays the memory requirements of our method. The biggest contribution is related to the fact that we need to load two images at any given time in order to extract the six-element vector of principal movements.

This points to the method’s applicability in real-time scenarios and the overall low resource requirements.

## 4. Discussion and Conclusions

We have developed a novel method for detecting falls using video data from standard cameras. Our technique is very robust—it does not depend on camera placement, which is very advantageous as often cameras are placed in various locations throughout a room. We provide two evaluations with which we can judge overall performance of the method—one on the 10% remainder of training data and another on an OOD set. From the evaluations, we see that it matches the accuracy of state-of-the-art optical flow methods such as the one presented in [18], without needing fusion of both video and audio data.

The method relies on a fast OF algorithm, which makes it very suitable for real-time problems such as the fall detection task. Both of these factors—freedom to place the camera in different locations and the computational efficiency—show us the universality of our method and its application for detection of falls in real-time.

GLORIA Net runs on a low to mid-range laptop CPU with low memory requirements and provides highly accurate results. Many contemporary methods for fall detection rely on processing through a dedicated GPU (both in training and analyzing new data). This is a key feature that demonstrates that it is competitive and practically relevant.

In addition to the above, it is worth mentioning that in the present work, we show an original approach to apply and use CNNs in video data. Most commonly, CNNs are used with images. A number of possible approaches [30,31] for application of Convolutional Neural networks in videos can be found in the literature, but the integration of the principal component OF parameters with a CNN is a novel contribution.

Recently, our group has introduced a system for real-time automated detection of epileptic seizures from video. Its capabilities were upgraded to include patient tracking and multiple camera coverage. GLORIA Net has been added to the existing system as it is modular in nature. We are currently observing its work, but so far, the addition has been very beneficial.

A limitation of our method is that it works with chunks of video data—collections of 150 frames from the video. This can be restricting in certain scenarios. As a future direction of development, we plan to increase responsiveness of the method so relevant parties can react quicker in the event of a fall. As with all other optical flow methods, GLORIA Net needs constant brightness of the scene in order to function properly. A future direction of work may include sensor fusion with an IR camera, utilizing the multi-spectral nature of the GLORIA technique. This would allow the fall detection method to function in more varied lighting conditions.

Another direction for future development is to define and extract features from the data that are related to the fall event in a physical sense. For example, we may look at the fall duration, the acceleration of the person, whether or not movement is periodic, and other similar quantities. Such preliminary refinement may help increase the accuracy of our fall detection scheme.

The datasets we have used in the current article are of videos in which a single person is present. We plan to test our method and upgrade it if needed on data that contains multiple moving people. A more thorough investigation of the additional LSTM block will be conducted on this data.

Many commercially available fall detection systems exist. Like the method here, they are described as accurate, inobtrusive, real-time, and automatic. What our system can do better is to rely on less sensors than wearable approaches [6,7]—it can work with a single camera while retaining real-time function with accurate results. In addition, cameras are readily available and can be found in many hospitals and homes, which removes the need to install and maintain special ambient sensors [5]. These benefits of our method, when compared to commercial products, further highlight its usefulness and ease of applicability.

## Figures and Tables

**Figure 1 sensors-25-05678-f001:**
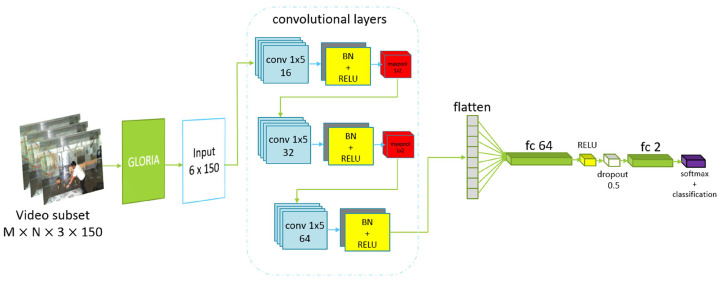
Description of the layers and architecture of GLORIA Net.

**Figure 2 sensors-25-05678-f002:**
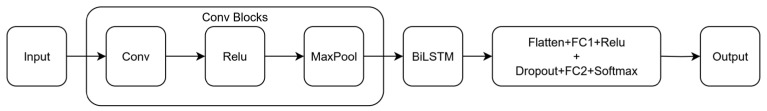
Addition of the LSTM block to our network.

**Figure 3 sensors-25-05678-f003:**
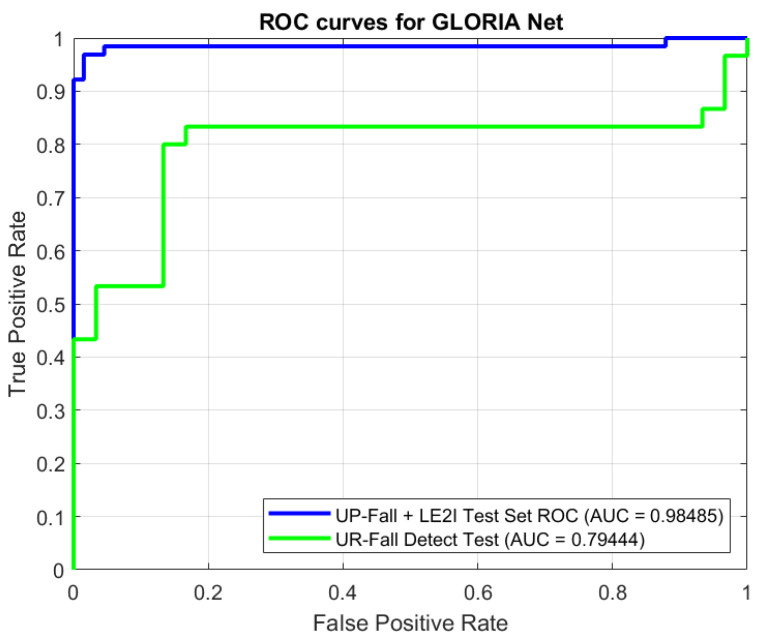
ROC curves for the GLORIA Net. Green line is for the performance on UR dataset, and blue line is for the performance on the UP-Fall+LE2I 10% subset. AUC values are provided in the legend.

**Figure 4 sensors-25-05678-f004:**
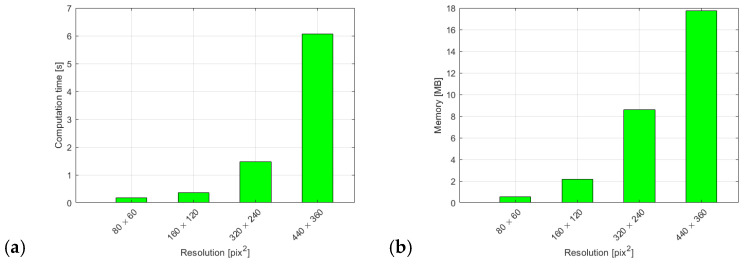
(**a**) Processing time required to make a classification for different frame resolutions. (**b**) Memory requirements for different frame resolutions.

**Table 1 sensors-25-05678-t001:** Accuracy, precision, sensitivity, specificity, and f1-score for both datasets.

Dataset	Accuracy	Sensitivity	Specificity	Precision	F1-Score
UP-Fall + LE2I	97.7%	98.1%	96.9%	97.0%	98.0%
UR	83.3%	83.3%	83.3%	83.3%	83.3%

**Table 2 sensors-25-05678-t002:** Accuracy, precision, sensitivity, specificity, and f1-score for UR dataset with and without the LSTM block.

Dataset	Accuracy	Sensitivity	Specificity	Precision	F1-Score
UR (+LSTM)	91.7% (↑ **8.4**%)	83.3%	100.0% (↑ **16.7**%)	100.0% (↑ **16.7**%)	90.9% (↑ **7.6**%)
UR (no LSTM)	83.3%	83.3%	83.3%	83.3%	83.3%

## Data Availability

Available upon request.

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
