# Peer review of "Automated Remote Detection of Falls Using Direct Reconstruction of Optical Flow Principal Motion Parameters"

_sensors, 2025, doi:10.3390/s25185678_

Round 1

Reviewer 1 Report

Comments and Suggestions for Authors

1. The introduction and related work sections require substantial improvement. The current literature review lacks comprehensive coverage of state-of-the-art fall detection methods and fails to position this work within the broader research landscape.

2. Dataset Selection and Justification: The authors should provide a thorough rationale for their dataset choice. Numerous datasets employing non-intrusive and privacy-preserving detection methods are available. The use of surveillance camera data raises significant privacy concerns for monitored subjects. The authors should address why alternative sensing modalities—such as egocentric cameras, wearable sensors, and bioradars—were not considered, as these offer comparable or superior performance while better preserving user privacy.

3. Presentation Quality: Figure 1 appears to be a low-quality screen capture and should be replaced with a properly formatted, publication-quality figure. Additionally, Tables 1 and 2 extend beyond the page margins and require reformatting to comply with journal guidelines.

4. Methodological Comparison: The experimental evaluation should include comparisons with current state-of-the-art algorithms. The proposed CNN-LSTM architecture represents a relatively basic approach that has been superseded by more sophisticated models in recent literature. The authors should justify their architectural choice and compare their method against contemporary approaches such as transformer-based models, graph neural networks, or advanced attention mechanisms to demonstrate the contribution of their work.

5. The section of results and discussion should be extended and improved.

Author Response

  1. The introduction and related work sections require substantial improvement. The current literature review lacks comprehensive coverage of state-of-the-art fall detection methods and fails to position this work within the broader research landscape.

We thank the reviewer for this comment. We have added several comments regarding the strong suits of our work in the Introduction chapter: L116-L124.

On the question of other works, we can say that there are many works on the topic of fall detection based on different CV algorithms. We have stated in P2_L88 that we will only go over other OF methods, as our method is also an OF technique. Going over other methods, based on non-OF techniques, would, in our view, shift the focus of our work. They are quite numerous, and a comprehensive discussion on many of them would be beyond our scope.

  1. Dataset Selection and Justification: The authors should provide a thorough rationale for their dataset choice. Numerous datasets employing non-intrusive and privacy-preserving detection methods are available. The use of surveillance camera data raises significant privacy concerns for monitored subjects. The authors should address why alternative sensing modalities—such as egocentric cameras, wearable sensors, and bioradars—were not considered, as these offer comparable or superior performance while better preserving user privacy.

We thank the reviewer for the thorough remark. We have chosen the three datasets in the current study for the following reasons:

  • They are video sets that contain footage of falls, which is the intended movement we wish to detect;
  • They are free, open, citable and available for direct download for the purposes of research;
  • They are labeled;
  • They are well known in the field of computer vision, on the topic of fall detections.

We have provided a long justification in the Introduction chapter as to why we chose to use a camera-based approach – L55 to L72. We will expand on this to address the reviewer’s comment: L72-L77.

We agree that user privacy is an essential aspect in the current context. That is one of the reasons that we use the GLORIA technique – it has substantial qualities when it comes to patient/person privacy. We have added comments that explain how the application of GLORIA helps in that regard: L116-L121. In citation [13] of the article, we provide more details on this topic.

  1. Presentation Quality: Figure 1 appears to be a low-quality screen capture and should be replaced with a properly formatted, publication-quality figure. Additionally, Tables 1 and 2 extend beyond the page margins and require reformatting to comply with journal guidelines.

We thank the reviewer for this note. We have reworked Figure 1 completely to address the mentioned issues. Both tables have also been reformatted.

  1. Methodological Comparison: The experimental evaluation should include comparisons with current state-of-the-art algorithms. The proposed CNN-LSTM architecture represents a relatively basic approach that has been superseded by more sophisticated models in recent literature. The authors should justify their architectural choice and compare their method against contemporary approaches such as transformer-based models, graph neural networks, or advanced attention mechanisms to demonstrate the contribution of their work.

We thank the reviewer for this comment, as it is a crucial moment for introducing new computer vision techniques. We do not have access to many of the current SOTA algorithms, and due to this, we cannot run tests and compare them to our method. With that in mind, we have chosen popular datasets that are easily accessible for everyone, as we cannot, in principle and in practice, run other techniques to test for performance. We can, however, use the popular and specialised fall detection video sets to benchmark our performance. If anyone wishes to compare results, they would be able to do so as the chosen video data is a staple in the topic of fall detection. That is the direction we have chosen to go in order to demonstrate detection capability.

As for the question of justification of architecture, we can summarize it in the following way: The GLORIA technique allows us to extract motion features from a subsection of a video. It allows us to extract principal motion information and organize it into a lower-dimensional structure – the 6x150 vector. The proposed neural architecture employs those vectors to classify the videos. CNN excels in extracting local tendencies based on which the LSTM cell captures the long-short-term dynamic of the input. Furthermore, such a network is compatible with a wide range of computational hardware as opposed to some transformer or graph-based variant that would be strongly dependent on GPU-specific access. We have added comments about this at the beginning of Chapter 2: L137-L144. We have also added more information about the LSTM approach: L217-L219.

  1. The section of results and discussion should be extended and improved.

We have added additional figures and text, both in the results and discussion sections, to address this comment. (Figure 1 and Figure 4, expanded comments in the last chapter)

Reviewer 2 Report

Comments and Suggestions for Authors

A new method for detecting and alerting falls based on Optical flow techniques is proposed in this work.  

The authors comprehensively highlight the aims of their research in the introduction; however, a broader discussion is needed on the computer vision and computational intelligence techniques currently used for fall detection and monitoring, and to specify the key features and performance limitations of these techniques.

Section 2 needs to be revised. What's missing is a presentation of the proposed method using an architectural diagram that allows us to understand its functional components and how they are interrelated. These functional components should be described separately, demonstrating their use within the framework.

Furthermore, an analysis of the method's computational complexity is needed to highlight its performance benefits in terms of execution time and memory management.

Conducting comparisons on just two datasets to verify the performance benefits of the LSTM approach may be insufficient. It would be necessary to extend the comparative tests to other datasets with different resolutions. The authors discuss this point, specifying whether they at least intend to conduct more extensive tests in future research.

What are the current performance limitations of the method? How can they be overcome in the future? This discussion should be included in the concluding section.

Author Response

  1. The authors comprehensively highlight the aims of their research in the introduction; however, a broader discussion is needed on the computer vision and computational intelligence techniques currently used for fall detection and monitoring, and to specify the key features and performance limitations of these techniques.

We thank the reviewer for this remark. We have added more comments in the introduction regarding the use of cameras for fall detection – L72-L77. We have also added several comments regarding the strong suits of our work in the Introduction chapter: L116-L124.

There are many works on the topic of fall detection based on different CV algorithms. We have stated in P2_L88 that we will only go over other OF methods, as our method is also an OF technique. Going over other techniques, based on non-OF techniques, would, in our view, shift the focus of our work. There are numerous, and a comprehensive discussion on many of them would be beyond our scope.

  1. Section 2 needs to be revised. What's missing is a presentation of the proposed method using an architectural diagram that allows us to understand its functional components and how they are interrelated. These functional components should be described separately, demonstrating their use within the framework.

We thank the reviewer for this note. We have reworked Figure 1, so that it adequately provides a full overview of the method and connectivity between different components.

We have also added more context at the beginning of Chapter 2: L137-L144.

  1. Furthermore, an analysis of the method's computational complexity is needed to highlight its performance benefits in terms of execution time and memory management.

We thank the reviewer for this note. Figure 4 provides detailed information regarding processing times for different resolutions. We add an additional subfigure that shows the memory requirements for the method. We remind that the method runs on the CPU (as opposed to other methods that are GPU-based with large VRAM requirements). Regarding the computational complexity of our neural architecture, we can say that it depends on the corresponding realisation. Still, for a fixed trained model, the complexity of the inference is linear with respect to the length of the input (i.e., the number of frames in the video).

  1. Conducting comparisons on just two datasets to verify the performance benefits of the LSTM approach may be insufficient. It would be necessary to extend the comparative tests to other datasets with different resolutions. The authors should discuss this point, specifying whether they at least intend to conduct more extensive tests in future research.

We thank the reviewer for the comment. We have expanded the Discussion chapter regarding the future direction of our work. We intend to include additional datasets with multiple moving (or falling) people. That would introduce new problems to solve, but it would extend the application of the current method.

  1. What are the current performance limitations of the method? How can they be overcome in the future? This discussion should be included in the concluding section.

We have extended the analysis in Figure 4 regarding memory requirements. We have added more context and comments in the Discussion section – L338-L341, L355-L359, L365-L368.

Reviewer 3 Report

Comments and Suggestions for Authors

In general, for each equation, the exact explanation of notations should be added. The more detailed explanations seem to be helpful for readers. There are many works done using video and please try to compare OF with other methods. There is a model to detect motions in video. Does OF give better performance than motion detection? It will be helpful to add more description for your network including LSTM in L201. The used method seems to straightforward. It is required to show the advantages of your approach.

P2_L82: It will helpful to add the disadvantages of using camera for fall detection.

P3_L127: Does the (x,y) coordinate include all pixels in the current frame?

P3_L132: Why the transformation of each color channel is calculated separately? Aren’t they all the same?

P3_L133: What Nc represents?

P3_L134: It will be better to decide to use t as time or frame number.

P4_L159: How the number of frames is decided as 150?

Author Response

  1. In general, for each equation, the exact explanation of notations should be added. The more detailed explanations seem to be helpful for readers. There are many works done using video, and please try to compare OF with other methods. There is a model to detect motions in video. Does OF give better performance than motion detection? It will be helpful to add more description for your network, including LSTM in L201. The used method seems to straightforward. It is required to show the advantages of your approach.

We thank the reviewer for this thorough comment. We have expanded the notation explanations. We have also added more detail to the LSTM approach in L217-L220. As all reviewers pointed out, we have expanded the discussion chapter to include more comments on the strengths and weaknesses of the proposed approach (L338-L341, L355-L359, L365-L368).

Numerous works and methods have been developed based on various CV algorithms. We cannot cover all of them in the introduction and provide a detailed comparison, as that would transform this work into a review article, which is not within its scope. We have explicitly stated in P2_L88 that we will only go over other OF methods, as our method is also an OF technique. Even today, OF methods still offer advantages to the SOTA computer vision techniques (transformers, correlation-based matching, end-to-end deep networks) in terms of processing speeds (where high-end GPUs are not available) and memory requirements.

  1. P2_L82: It will helpful to add the disadvantages of using camera for fall detection.

We thank the reviewer for this note. It is entirely correct, and we have added a summary of the disadvantages related to camera use for fall detection (L72-L77).

  1. P3_L127: Does the (x,y) coordinate include all pixels in the current frame?

Yes, by denoting the current frame as Lc(x,y,t), we mean the entire image at time t. (added)

  1. P3_L132: Why the transformation of each color channel is calculated separately? Aren’t they all the same?

Traditionally, OF methods use the greyscale value of the images. GLORIA is a multi-channel method that can take advantage of RGB images (three color channels vs. one intensity channel in greyscale). Please note that the velocity field v in equation (1) at P4_L152 is common for all spectral channels. In this way, all the information simultaneously from all channels provides an early binding, data fusion approach. OF reconstruction is, in general, and especially in the case of using only intensity images, an underdetermined problem because the local velocities in the directions of constant intensity can be arbitrary.  This degeneracy is less likely to occur in multichannel images.  It is a beneficial property of the GLORIA method, as it reduces possible degeneracies (zero gradient in flat intensity regions) in the inverse problem, provides more constraints per pixel, is less sensitive to illumination changes, and accounts for the natural image structure. We added the above explanation to the revised text.

  1. P3_L133: What Nc represents?

We thank the reviewer for this remark. Nc is the total number of colour channels. We have added this to the text.

  1. P3_L134: It will be better to decide to use t as a time or frame number.

We thank the reviewer for this note. We have added a clarification in the text to explain what the parameter 't' means - time.

  1. P4_L159: How the number of frames is decided as 150?

The number of frames is decided based on two factors. The first is related to the videos we have used for training and testing. Most of them are in the duration range of 120-180 frames. Taking this value of 150 frames ensures a standardized duration and subsequent input vector size for our CNN. The second reason is the physical nature of the fall. Based on our experience reviewing fall data, a fall event lasts for 1-2.5 seconds. We want to take a sufficiently long interval of time (5-8 seconds), so that the motion related to the fall is placed well within the current subsection of the video. We want to avoid overlapping sliding windows.

Round 2

Reviewer 2 Report

Comments and Suggestions for Authors

The authors have revised their manuscript, incorporating all my suggestions. I consider this paper publishable in its current version.

Reviewer 3 Report

Comments and Suggestions for Authors

The questions are well answered and applied in the revised paper. Minor modification is well applied.